# Clinical Presentation, Treatment Outcomes, and Demographic Trends in Vestibular Schwannomas: A 135-Case Retrospective Study

**DOI:** 10.3390/jcm14020482

**Published:** 2025-01-14

**Authors:** Corneliu Toader, Petrinel Mugurel Rădoi, Milena-Monica Ilie, Razvan-Adrian Covache-Busuioc, Vlad Buica, Luca-Andrei Glavan, Matei Serban, Antonio Daniel Corlatescu, Carla Crivoi, Radu Mircea Gorgan

**Affiliations:** 1Department of Neurosurgery, “Carol Davila” University of Medicine and Pharmacy, 020021 Bucharest, Romania; corneliu.toader@umfcd.ro (C.T.); milena-monica.ilie0720@stud.umfcd.ro (M.-M.I.); razvan-adrian.covache-busuioc0720@stud.umfcd.ro (R.-A.C.-B.); vlad.buica0720@stud.umfcd.ro (V.B.); luca-andrei.glavan0720@stud.umfcd.ro (L.-A.G.); matei.serban2021@stud.umfcd.ro (M.S.); antonio.corlatescu0920@stud.umfcd.ro (A.D.C.); radu.gorgam@umfcd.ro (R.M.G.); 2Department of Vascular Neurosurgery, National Institute of Neurology and Neurovascular Diseases, 077160 Bucharest, Romania; 3Faculty of Mathematics and Computer Science, University of Bucharest, 050663 Bucharest, Romania; crivoicarla02@gmail.com; 4Department of Neurosurgery, Bagdasar-Arseni Clinical Emergency Hospital, 041915 Bucharest, Romania

**Keywords:** vestibular schwannoma, hydrocephalus, clinical outcomes, vestibulocochlear nerve, hearing impairment, facial palsy, balance disorders, visual impairment, gamma knife radiosurgery, open surgery

## Abstract

**Background**: This study presents a comprehensive analysis of 135 cases of vestibular schwannoma (VS) treated between 2006 and 2022 at the National Institute of Neurology and Neurovascular Diseases in Bucharest, Romania. The investigation focuses on the clinical presentation, treatment outcomes, and demographic trends of VS patients, highlighting region-specific insights that fill critical gaps in Eastern European data. **Methods**: Patients were treated with either open surgery (93.3%) or gamma knife radiosurgery (6.6%). The study identifies predominant symptoms, including hearing impairment, facial palsy, and balance disorders, with variations observed across age and gender subgroups. Comorbidities such as hypertension and obesity were prevalent, and they influenced perioperative risks. **Results**: Post-treatment outcomes showed a significant correlation between clinical symptoms and treatment modalities, with a majority achieving favorable results. The findings emphasize the need for tailored approaches in VS management and underscore the importance of region-specific factors in influencing clinical outcomes. **Conclusions**: This study contributes to refining treatment strategies and improving healthcare delivery for VS patients in Romania and beyond.

## 1. Introduction

Schwannomas represent tumors of a benign nature that contain a capsule and have a variety of locations such as spinal nerve roots, internal organs, or soft tissues. The majority of cases are sporadic, with 10% of schwannomas being linked to familial neurofibromatosis type 2 (NF2). A frequently affected cranial nerve is the vestibulocochlear nerve in the vestibular section [1]. NF^2^ is the primary genetic condition associated with the development and recurrence of vestibular schwannomas, typically presenting with bilateral tumor growth [2,3]. The prevalence of NF2 is estimated at 1 in 25,000 to 1 in 33,000 in Europe, with similar rates likely in Romania; however, comprehensive data on NF2 incidence in Romania are lacking. Due to constraints in genetic testing, particularly in resource-limited settings, data on NF2 incidence are sparse, highlighting the need for further investigation to understand its impact on VS demographics and outcomes in Romanian patients.

VSs, also known as acoustic neurinomas, consist of benign tumors originating from Schwann cells that affect the auditory and vestibular function of the eighth cranial nerve. They comprise 8% of all intracranial tumors, with an estimated annual incidence of 1 in 100,000 persons [4,5]. It is estimated that the actual incidence of VS is higher due to more advanced diagnostic techniques available at present. The literature approximates that 85% of all pontocerebellar angle tumors are VS [6]. Over the past four decades, the age of diagnosis for vestibular schwannomas has shifted from 49 years in 1976 to approximately 60 years today [7]. However, patients presenting with NF2 develop the disease earlier in life and have a higher chance of bilateral lesions [8,9].

The most prevalent symptoms presented by patients with VS are of an otologic nature and consist of progressive hearing loss and tinnitus (approximately 60%). The advancement of the disease leads to neurologic manifestations because of brainstem compression, resulting in facial palsy, vertigo, headache, and hydrocephalus [10,11]. For the diagnosis of VS, MRI is considered the gold standard for patients presenting with persistent hearing loss or tinnitus. It is important to note that the difference in sensitivity and specificity between gadolinium-enhanced T1 and non-contrast T2 scans is negligible [10,12].

Risk factors associated with VS remain controversial due to an insufficient number of significant studies necessary for confirmation. Corona et al. analyzed a systematic review of 20 papers from the literature that presented different risk factors which could be correlated with VS. They considered education, family income, occupation, hay fever, exposure to ionizing and non-ionizing radiation, and exposure to high noise levels as possible risk factors, but further studies are required [13]. Other more recent studies focus on the impact of radiation produced by mobile phones, hay fever, asthma, and tobacco usage [14,15].

Treatment options for VS consist of conservative management, surgical resection, and radiotherapy, all with advantages and disadvantages. The first option is accepted because of the slow progression rate of the tumor. There is no supplementary risk in mortality for patients with delayed surgical intervention [16]. There are various surgical approaches to VS which are indicated for the excision of larger masses, including the middle fossa, retrosigmoid, transotic, or translabyrinthine. The last procedure, radiotherapy, is suitable for tumors of smaller dimensions because the purpose of this technique is to minimize the growth of VS [10,17].

For this article, we have collected 135 cases of patients presenting with vestibular schwannomas that underwent surgery or gamma knife radiosurgery in our clinic over the span of 16 years. While the clinical presentation and surgical treatment options for VS are well established globally, there is a scarcity of region-specific data from Romania. Although existing studies provide general outcomes on VS treatment, including gamma knife radiosurgery and microsurgical resection, limited data focus on the Romanian population [18]. Gaps remain in understanding whether regional factors, such as access to healthcare, socioeconomic differences, and patient demographics, influence treatment outcomes. Furthermore, studies on long-term post-surgical outcomes and specific treatment challenges in Romania are lacking. Addressing these gaps can provide insights into population-specific variations in VS treatment and contribute to refining approaches to care.

## 2. Materials and Methods

This study examines 135 patient cases treated with either open surgery or gamma knife radiosurgery between 2006 and 2022 at the Department of Neurosurgery, National Institute of Neurology and Neurovascular Diseases in Bucharest, Romania. Statistical analysis and figure plotting were performed using Python version 3.10, developed by the Python Software Foundation (9450 SW Gemini Dr., ECM# 90772, Beaverton, OR, USA). The analysis utilized Python libraries such as pandas, numpy, seaborn, and matplotlib. Of these, 126 cases involved open surgical intervention. A thorough review was conducted on the preoperative, intraoperative, and postoperative stages of patient management, focusing on variables such as age distribution, tumor localization, facial palsy, hearing impairments, and visual impairments. Postoperative complications and the number of reoperations were also discussed. Exclusion criteria included patients with incomplete medical records, those who had received treatment elsewhere prior to admission, and patients lacking documented follow-up. A flowchart (Figure 1) illustrates the study design, highlighting the selection process and patient distribution. 

This study was conducted in accordance with the principles outlined in the Declaration of Helsinki and received ethical clearance from the National Institute of Neurology and Neurovascular Diseases in Bucharest, Romania (Ethical Review Board of the National Institute of Neurology and Neurovascular Diseases; 7653). Relevant clinical data, including pre- and postoperative symptoms, tumor localization, and treatment modalities, were collected from the patients’ files. All data were processed in compliance with current GDPR regulations, and informed consent was obtained from every participant.

Statistical analyses and figure generation were carried out using Python 3.10, which was developed by the Python Software Foundation (9450 SW Gemini Dr., ECM# 90772, Beaverton, OR 97008, USA). Key Python libraries utilized for data handling and visualization included pandas, numpy, seaborn, and matplotlib.

## 3. Results

### 3.1. Patient Demographics

Patient demographics, with a focus on age and gender, were examined. Of the 135 patients, 38.5% (n = 52) were male and 61.5% (n = 83) were female, resulting in a male-to-female ratio of approximately 1:1.6.

The age of patients at presentation with VS ranged from 19 to 83 years, with a mean age of 52.02 years (SD = 14.62), indicating a moderate variability in the cohort’s age distribution. The median age was 54 years, which closely aligns with the mean, suggesting a relatively symmetrical distribution of ages. The highest prevalence of VS was observed in the 55- to 64-year age group, accounting for 31.2% of patients, followed by the 45- to 54-year age group, representing 19.2% of the cohort (Figure 2). A *t*-test was performed to compare the observed age distribution against an expected population distribution. The test yielded a result of *p* = 1.00, indicating no statistically significant deviation and confirming that the age distribution in this cohort is consistent with expected variations in a general population with VS.

Table 1 presents a detailed analysis of symptom prevalence and surgical approaches in our cohort of 135 patients diagnosed with vestibular schwannoma. By stratifying the data into demographic subgroups, gender (male and female) and age (young < 40 years, middle-aged 40–59 years, elderly ≥ 60 years), we aim to provide insights into the distribution of VS-related symptoms and the choices of surgical management.

The prevalence of primary symptoms, including hearing impairment, facial palsy, balance disorders, visual impairment, and hydrocephalus, is displayed for each subgroup, allowing for comparative insights. Surgical approaches, primarily open surgery and gamma knife radiosurgery, are also stratified by demographics to illustrate treatment preferences or considerations based on patient characteristics.

This breakdown highlights the variations in clinical presentation and outcomes by demographic factors, providing a comprehensive overview of VS’s impact and treatment efficacy across age and gender groups within this cohort.

### 3.2. Presentation

#### 3.2.1. Glasgow Coma Scale

Upon admission, the Glasgow Coma Scale was used to evaluate each patient’s mental status. The majority (97%, n = 131) had mild brain injuries with scores of 13 or above, 43.3% (n = 58) scored 15, 39.6% (n = 53) scored 14, and 14.9% (n = 20) scored 13 (Figure 3). An additional 1.5% (n = two) presented with moderate brain injuries (GCS score of twelve), while only one patient arrived with a GCS score of three, indicating severe brain injury. In this case, the comatose state was primarily due to obstructive hydrocephalus, exacerbated by significant comorbidities (stage 2 obesity and stage 2 hypertension).

#### 3.2.2. Clinical Symptoms

The symptoms exhibited by our patients were significant and played a crucial role in diagnosing VS. These symptoms span the fields of otorhinolaryngology, ophthalmology, and neurology, which will be discussed in detail below.

#### 3.2.3. Hearing Impairment

Hypoacusis was a prevalent symptom found in our analysis (Figure 4), affecting 68.8% (n = 93) of the patients. A total of 40% (n = 53) presented with right hypoacusis, while 28.8% (n = 39) presented with left hypoacusis. A significant difference in the affection of the left ear compared to the right one has not been established, with a L:R ratio of 1:1.38. A total of 2.2% (n = three) of patients presented with unilateral anacusis. One person presented with bilateral hypoacusis.

#### 3.2.4. Facial Palsy

The distribution of peripheral facial palsy between the left and right sides showed no substantial difference, mirroring the pattern observed with hypoacusis. The left-to-right ratio for facial palsy occurrence was 1:1.22. A total of 17.8% (n = 25) of patients did not exhibit facial palsy.

A significant correlation was found between peripheral facial palsy and hearing loss on the same side, with 60% of patients (n = 81) experiencing both symptoms concurrently (Figure 5).

#### 3.2.5. Dysphagia

Dysphagia was not as prevalent as hypoacusis and facial palsy, with only 13.3% (n = 18) of patients being affected.

#### 3.2.6. Unsystematized Balance Disorders

A total of 91.8% (n = 124) of patients presented with vestibular afflictions, leading to gait and balance disorders, while only a small fraction, specifically 8.1% (n = 11), showed no such afflictions (Figure 6).

#### 3.2.7. Visual Impairment

In the study, 16.2% (n = 22) of patients presented with at least one visual impairment, including diplopia, nystagmus, blurred vision, photophobia, visual field deficits, visual acuity reduction, and cecity (Figure 7). Notably, these symptoms were either unilateral or bilateral.

#### 3.2.8. Hydrocephalus

While a significant portion of patients with VS did not have hydrocephalus upon admission, approximately 33.3% (n = 45) did experience some form of the condition. Obstructive internal hydrocephalus was particularly prevalent, affecting 29.6% (n = 30 patients), making it the most common type among these patients (Figure 8). Normal pressure hydrocephalus accounted for 2.2% (three patients), while 1.5% (two patients) were diagnosed with active hydrocephalus, which required immediate surgical intervention.

### 3.3. Risk Factors

In examining comorbidities within our patient cohort, obesity, arterial hypertension, and diabetes mellitus were identified as the most prevalent health conditions, warranting a detailed analysis due to their potential impact on treatment outcomes and overall patient management. The majority of patients fell within a normal or overweight body mass index (BMI) range, with 91.8% exhibiting a BMI below 30 kg/m^2^. However, 11 patients presented with obesity, characterized by a BMI exceeding 30 kg/m^2^. Of these, nine had a BMI between 35 and 39.9 kg/m^2^, and two patients were classified with morbid obesity (BMI > 40 kg/m^2^). Obesity is a known factor in surgical complexity, influencing anesthetic risk and postoperative recovery. Its presence in our cohort underscores the importance of tailored perioperative care for VS patients with elevated BMI.

Arterial hypertension was notably the most common comorbidity, affecting 24.4% (n = 33) of patients. Among these, hypertension severity varied: eight patients presented with grade 1 hypertension, twenty-two with grade 2 hypertension, and three experienced a hypertensive crisis (blood pressure > 180/120 mmHg). Given that hypertension can increase perioperative risks, especially in neurosurgical cases, its high prevalence among VS patients highlights the need for careful blood pressure management before, during, and after surgery to mitigate potential complications.

Diabetes mellitus, though less common, was present in 3.7% (n = five) of patients, and all were diagnosed with type 2 diabetes. Diabetes is associated with increased risks of surgical site infections, delayed wound healing, and neuropathy, which can complicate postoperative outcomes. Although obesity, hypertension, and diabetes are not direct risk factors for VS, their prevalence within this cohort emphasizes the importance of comprehensive health assessments and individualized perioperative management strategies. By understanding these comorbidities, clinicians can better anticipate and address complications, ultimately improving care quality and outcomes for VS patients.

### 3.4. Location

In our cohort of 135 patients with vestibular schwannoma, the majority of tumors were located in the cerebellopontine angle (CPA). Specifically, 51.1% (n = 69) were situated in the right CPA, while 42.2% (n = 57) were found in the left CPA. Additionally, 1.5% (n = two) of tumors were located in the right internal auditory meatus, and another 1.5% (n = two) were found in the left internal auditory meatus (Figure 9). In five cases (3.7%), the precise localization could not be accurately determined, leading to their classification as “Posterior cranial fossa-unspecified”.

### 3.5. Treatment

The study involved patients who received two different treatments: a total of 93.3% (one hundred and twenty-six patients) had surgical removal of the tumor, while 6.6% (nine patients) were treated with a single session of gamma knife radiosurgery at a dose of 12 Gy.

### 3.6. Patients’ Outcome

The outcome of our patients was assessed using the Glasgow Outcome Scale, which evaluated the mental state on a scale from 1 to 5 at discharge. The majority of our patients had a positive outcome, dictated by a score of 5–65.1% (n = eighty-eight), followed by 31.1% (n = forty-two) of patients with a score of 4. A minority (n = three) had a score of three. Only two in-hospital deaths were acknowledged. Postoperative assessments suggest that hearing impairment persisted for a significant number of patients, consistent with studies that noted the difficulty of preserving hearing in vestibular schwannoma cases, especially those requiring extensive resection. Successful hearing preservation post-surgery is closely influenced by factors such as tumor size, precise location, and the surgical approach chosen.

#### 3.6.1. Remission of Facial Palsy

After treatment, 24.4% (n = 33 patients) fully recovered from facial palsy. Partial recovery of facial motor function was seen in 3.7% (n = five patients), while the majority, 54.8% (n = seventy-four patients), continued to have dysfunction. The remaining 17% (n = 23 patients) did not have facial palsy when they were admitted (Figure 10).

#### 3.6.2. Remission of Hyodrcephalus

After treatment, hydrocephalus completely resolved in 17.8% (n = 24) of patients. Partial improvement was seen in 4.4% (n = six) of cases, while 11.1% (n = fifteen) of patients showed no improvement. The majority of patients, 66.7% (n = 90), did not have hydrocephalus when they were first examined (Figure 11).

### 3.7. Relapse

All patients treated for vestibular schwannoma were monitored for at least 2 years. Approximately 86.6% (n = 117) of the patients had a favorable outcome with no relapses, with an overall relapse rate of 13.3%. Among the nine patients who received gamma knife Radiosurgery, two experienced a relapse during the follow-up period. In contrast, out of the 126 patients who underwent surgical removal of their tumors, 16 experienced relapses. It is important to note that these relapses occurred either within the 2-year follow-up period or later, depending on the frequency of the patients’ routine check-ups and the year they were operated on. The earliest relapse was observed 8 months after treatment, while the most distant relapse occurred 156 months post-treatment.

## 4. Discussion

In our study, 61.5% of patients were women, resulting in a male-to-female ratio of 1:1.6. A chi-square test was performed to evaluate the gender distribution, revealing a statistically significant deviation from equality (*p* = 0.0076). This ratio raises the ongoing question in the literature about the potential genetic, in addition to environmental, factors that influence this balance. Numerous studies that have delved into the biochemical and immunohistochemical aspects of vestibular schwannomas have shown that vestibular schwannomas can express estrogen receptors, progesterone receptors, or both [19,20,21]. These findings are clinically significant, highlighting the potential of endocrinologic therapies as adjunct treatments for cases with postoperative residual vestibular schwannomas [22]. Monsell et al.’s study found no correlation between gender and the prevalence of vestibular schwannomas with estrogen receptors [19]. Further research is needed to better understand the demographic profiles of patients with VS and to explore new therapeutic approaches, including anti-progesterone and anti-estrogen therapies.

In a study by Reznitsky et al. involving 3637 patients diagnosed with sporadic/unilateral vestibular schwannomas over a 40-year span from 1976 to 2015, notable shifts in the age at diagnosis were observed. The median age at diagnosis rose from 49.2 years in 1976 to 60 years in 2015 [7]. In our study, which included 135 cases from 2006 to 2022, the overall median age at VS diagnosis was 54 years old, with the youngest diagnosis age in 2020 (Figure 12). This was likely due to a period of underdiagnosis in the general population, similar to what was observed with malignant tumors, as the COVID-19 pandemic in 2020 significantly limited access to healthcare [23].

In patients with VS and obstructive hydrocephalus leading to elevated intracranial pressure, papilledema is typically the cause of visual impairments. However, the literature raises another possible pathway in which patients with VS progress to visual deficits, specifically without a strong causation or correlation to the abnormal accumulation of cerebrospinal fluid (CSF) in the ventricles [24,25].

Interestingly, one study by Mishra et al. has found that female gender and younger age might be risk factors for developing visual deficits even in the absence of hydrocephalus [26]. This highlights the need for a more comprehensive understanding of visual complications in VS patients beyond the traditional focus on hydrocephalus-related causes. In our case, of the 22 patients presenting with visual deficits, 13 had either obstructive or active hydrocephalus. The remaining nine patients showed no imaging evidence or symptoms indicative of abnormal CSF accumulation, suggesting that visual deficits in vestibular schwannoma can occur through mechanisms unrelated to hydrocephalus. Among those with hydrocephalus and visual deficits, there were five males (ages 33, 41, 54, 57, and 64) and four females (ages 31, 35, 49, and 54).

A significant correlation was found between peripheral facial palsy and hearing loss on the same side, with 60% of patients (n = 81) experiencing both symptoms concurrently. Of these eighty-one patients, seventy-seven underwent open surgery, while only four were suitable for stereotactic radiosurgery. This suggests that the coexistence of facial palsy and hearing impairments may often necessitate open surgery, though further investigation is needed to confirm this trend.

A study by Goshtasbi et al. found that postoperative myocardial infarction and wound infection were more commonly observed in obese patients (BMI ≥ 30) undergoing surgery for vestibular schwannomas. These complications occurred at statistically significant rates in the obese group compared to non-obese patients [27]. In our study, we observed a similar pattern. One of the two in-hospital deaths occurred in a patient with stage 2 obesity. Following open surgery, this patient’s health deteriorated significantly, leading to multiple complications, including a myocardial infarction.

As stated before, at our institute, open surgery was the primary treatment modality for VS, accounting for 93.3% (n = 126) of cases. A chi-square test was performed to evaluate the distribution of treatment modalities, confirming a statistically significant preference for open surgery over gamma knife radiosurgery (*p* < 0.05).

Among these, 118 tumors located in either the left or right pontocerebellar angles were addressed using the retrosigmoid (lateral suboccipital) approach, with the goal of preserving facial nerve integrity. A systematic review by Ansari et al. found that the retrosigmoid approach is the most versatile for preserving facial nerve function across various tumor sizes, although it is associated with a higher risk of postoperative pain [28]. These findings further support our decision to favor the retrosigmoid approach over the middle fossa approach. Open surgery was also performed in three cases involving the internal auditory meatus and in five cases where the precise tumor location could not be determined from the medical records. One of the key indications for tumor resection was the presence of hydrocephalus, particularly obstructive or active hydrocephalus.

In contrast, gamma knife radiosurgery was selected for a smaller proportion of patients, specifically 6.66% (n = nine). Among these, eight tumors were located in the pontocerebellar angle, and one was in the left internal auditory meatus. This treatment option was particularly considered for older patients, as vestibular schwannoma resection in this demographic is associated with a higher mortality rate [27]. The decision on gamma knife radiosurgery was influenced by several patient-specific factors, including the absence of hydrocephalus or the presence of normal pressure hydrocephalus, and the absence of visual impairments. These criteria were important in determining the suitability of gamma knife radiosurgery as an alternative to open surgery for these patients. Additionally, patient refusal of open surgery accounted for the three cases where the vestibular schwannoma’s maximum diameter exceeded 3.5 cm.

The Koos classification for vestibular schwannomas was first introduced in 1998, categorizing tumors based on their extension beyond the internal auditory canal and the degree of brainstem compression [29]. Although Erickson et al. found that different specialists can reliably apply this classification when analyzing various magnetic resonance imaging (MRI) scans, there is limited research on its effectiveness in guiding specific surgical approaches or predicting outcomes [30]. Furthermore, another study by Erickson et al. could not establish a link between the Koos classification and postoperative facial nerve function [31]. Although the Koos classification is recognized in the field, its broad limits restrict its clinical applicability, which is why we did not heavily emphasize it in our study.

## 5. Limitations

The limitations of our study include several critical gaps that impact the generalizability and depth of our findings. First, we lacked data on tobacco and alcohol consumption among patients, which are lifestyle factors known to potentially influence the risk and progression of VS. This absence restricts our ability to fully assess environmental contributors to VS within our cohort. Additionally, while the Koos grading scale is commonly used to guide treatment planning in VS, we did not incorporate it into our decision-making process as its predictive utility remains limited in certain contexts, and it is not universally applied in Romanian clinical settings.

Another limitation is the absence of genetic testing for NF2 due to socioeconomic constraints. NF2 is a key genetic factor associated with bilateral VS and could influence relapse rates; however, without this data, our study cannot draw definitive conclusions regarding the relationship between treatment outcomes and genetic predispositions for VS recurrence. This limitation underscores a broader gap in Southeastern European healthcare, where resource limitations hinder comprehensive genetic profiling for VS patients. Incorporating NF2 testing in future studies would provide a clearer understanding of hereditary VS in this region.

Finally, this study’s retrospective design, covering a 15-year period, may introduce temporal biases due to advances in surgical techniques, imaging, and postoperative care, all of which can impact treatment outcomes. This is particularly pertinent in Romania and Southeastern Europe, where healthcare resources and access vary considerably. Prospective studies with standardized, contemporaneous data collection methods are recommended to accurately reflect current treatment environments and to facilitate meaningful comparisons with data from other regions.

The scarcity of comprehensive data on genetic, lifestyle, and healthcare factors in Southeastern Europe presents a challenge to fully understanding VS outcomes in this population. Addressing these limitations in future research will help refine treatment strategies and provide more population-specific insights into the management of VS in this region.

## 6. Conclusions

Our study provides valuable insights into the clinical presentation, treatment outcomes, and potential risk factors associated with vestibular schwannomas, with a unique focus on a Romanian cohort, addressing a critical gap in region-specific data from Eastern Europe. This study offers one of the first detailed analyses of how local factors, such as healthcare access and socioeconomic differences, may influence VS outcomes in this region. Considering alternative treatment approaches like conservative management and stereotactic radiosurgery, our findings support open surgery for particular cases, especially given the cohort’s observed male-to-female ratio of 1:1.6. Additionally, this study underscores the need for further investigation into the role of endocrinological therapies, such as anti-estrogen and anti-progesterone treatments, which could refine strategies to improve patient outcomes in VS management.

## Figures and Tables

**Figure 1 jcm-14-00482-f001:**
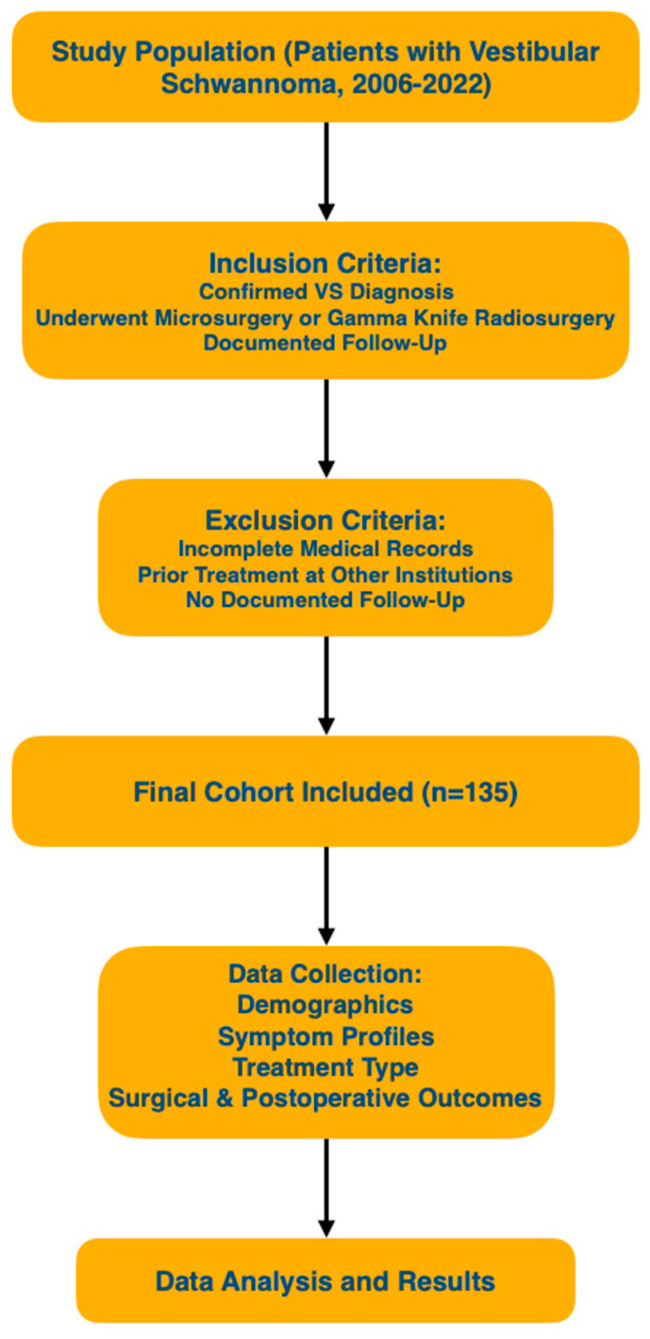
This flowchart outlines the study design for vestibular schwannoma patients, including inclusion and exclusion criteria, final cohort selection, data collection points, and the progression to data analysis. It provides a clear visual overview of the study methodology from patient selection to outcome analysis.

**Figure 2 jcm-14-00482-f002:**
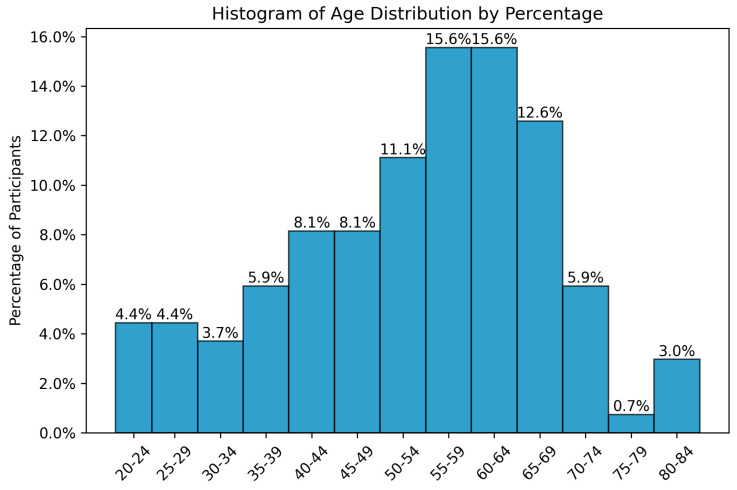
Age distribution of vestibular schwannoma cases in 15-year intervals in a 135-case retrospective study.

**Figure 3 jcm-14-00482-f003:**
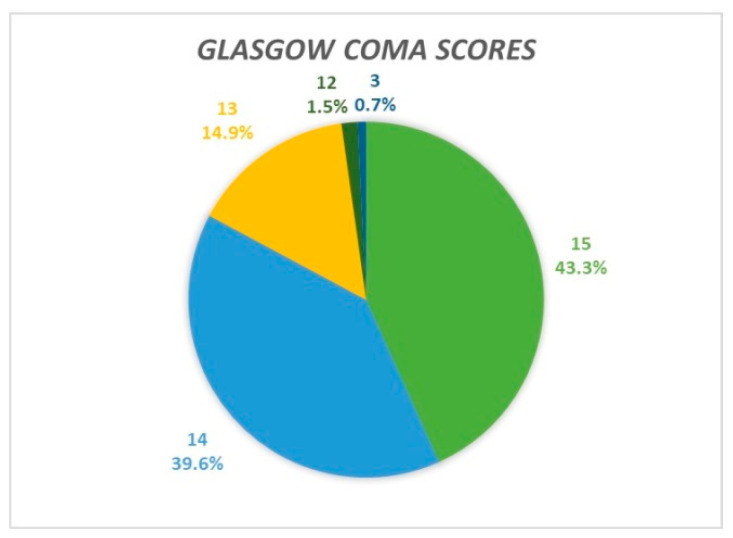
Percentage distribution of Glasgow Coma Scores upon patient admission.

**Figure 4 jcm-14-00482-f004:**
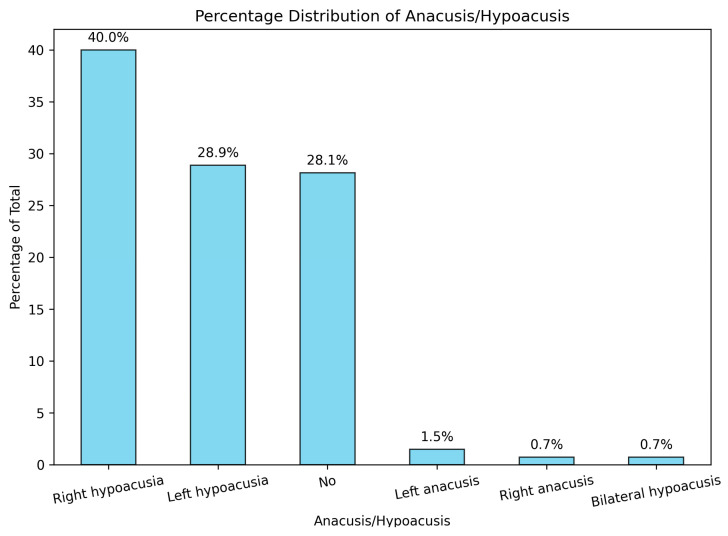
Percentage distribution of hypoacusis and anacusis upon admission.

**Figure 5 jcm-14-00482-f005:**
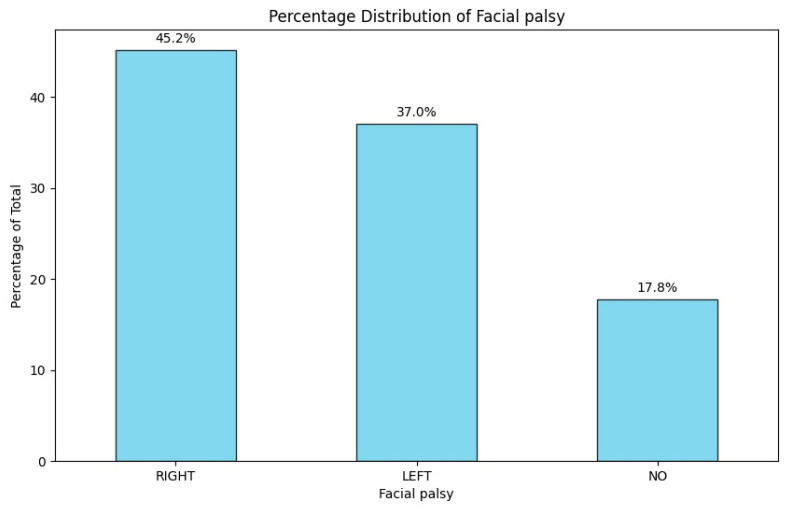
Percentage distribution of facial palsy upon patient admission.

**Figure 6 jcm-14-00482-f006:**
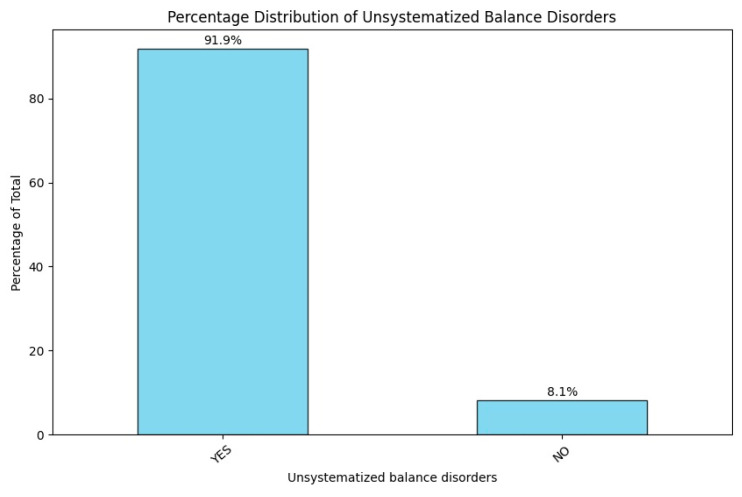
Percentage distribution of unsystematized balance disorders upon patient admission.

**Figure 7 jcm-14-00482-f007:**
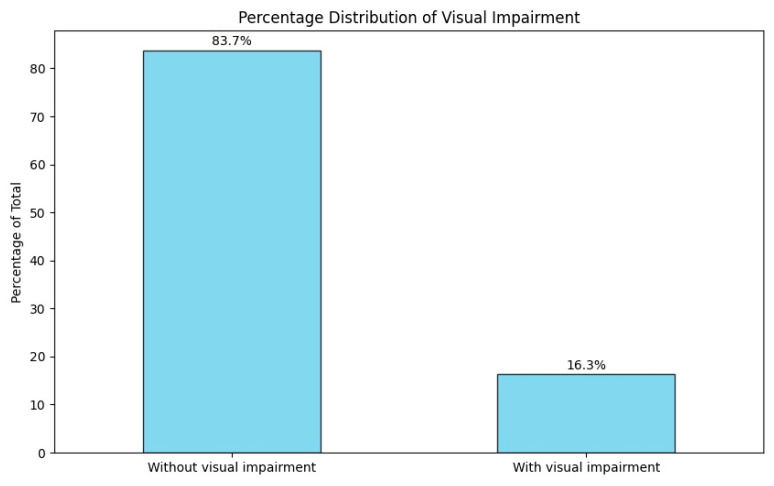
Percentage distribution of visual impairments upon patient admission.

**Figure 8 jcm-14-00482-f008:**
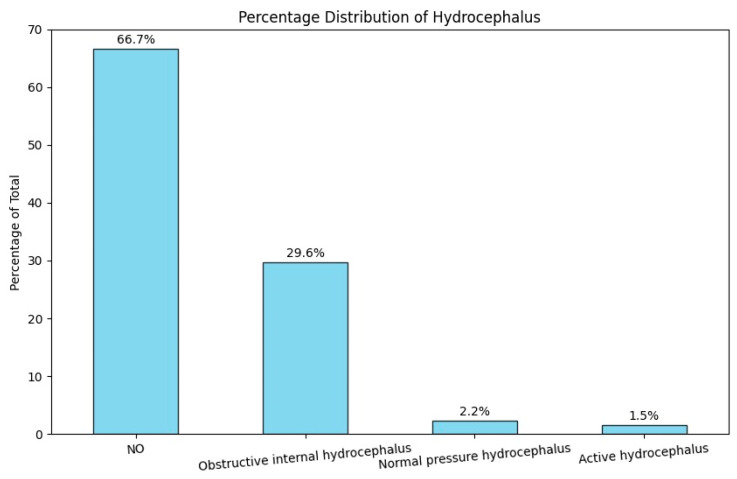
Percentage distribution of hydrocephalus upon patient admission.

**Figure 9 jcm-14-00482-f009:**
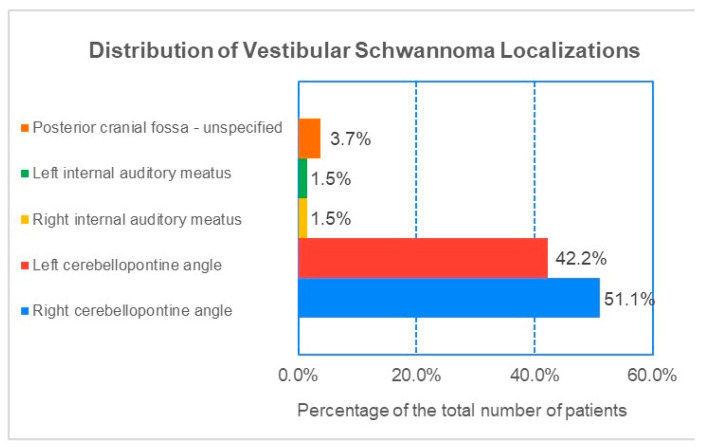
Percentage distribution of VS localizations in our cohort.

**Figure 10 jcm-14-00482-f010:**
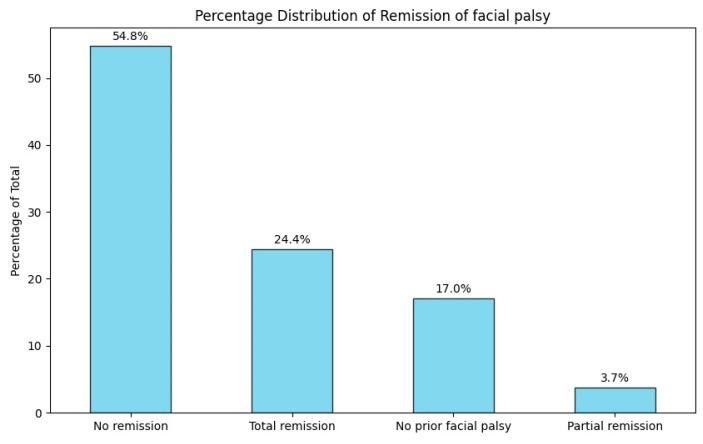
Percentage distribution of remission of facial palsy.

**Figure 11 jcm-14-00482-f011:**
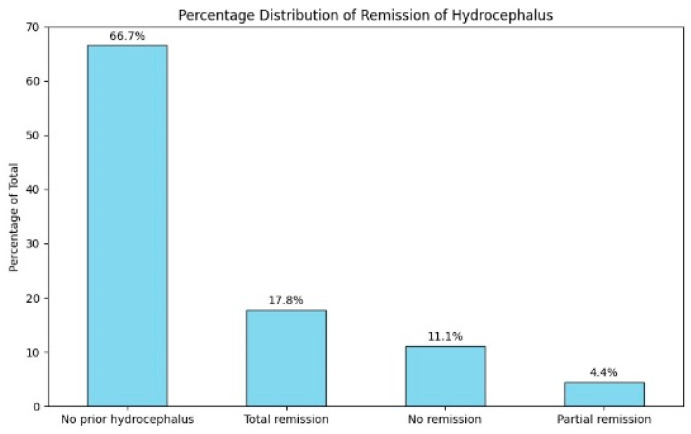
Percentage distribution of remission of hydrocephalus.

**Figure 12 jcm-14-00482-f012:**
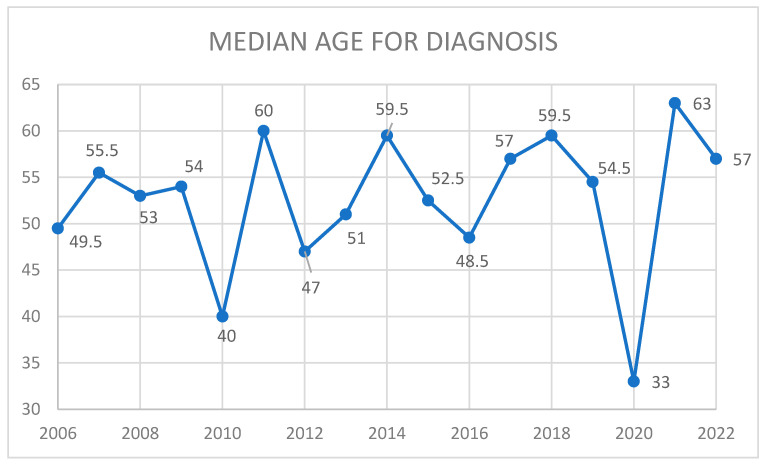
Median age for diagnosis of VS in our cohort for every year between 2006–2022.

**Table 1 jcm-14-00482-t001:** This table presents demographic and clinical data on vestibular schwannoma patients by gender and age group.

Category	Total (n = 135)	Male (n = 52)	Female (n = 83)	Young (<40 Years, n = 20)	Middle-Aged (40–59 Years, n = 60)	Elderly (≥60 Years, n = 55)
Symptom Prevalence						
Hearing Impairment (%)	68.8% (n = 93)	65.4% (n = 34)	71.1% (n = 59)	60.0% (n = 12)	70.0% (n = 42)	72.7% (n = 40)
Facial Palsy (%)	82.2% (n = 111)	80.8% (n = 42)	83.1% (n = 69)	75.0% (n = 15)	85.0% (n = 51)	81.8% (n = 45)
Balance Disorders (%)	91.8% (n = 124)	88.5% (n = 46)	93.9% (n = 78)	90.0% (n = 18)	91.7% (n = 55)	92.7% (n = 51)
Visual Impairment (%)	16.2% (n = 22)	13.5% (n = 7)	18.1% (n = 15)	10.0% (n = 2)	15.0% (n = 9)	20.0% (n = 11)
Hydrocephalus (%)	33.3% (n = 45)	30.8% (n = 16)	34.9% (n = 29)	25.0% (n = 5)	31.7% (n = 19)	38.2% (n = 21)
Surgical Approach						
Open Surgery (%)	93.3% (n = 126)	92.3% (n = 48)	93.9% (n = 78)	90.0% (n = 18)	93.3% (n = 56)	96.4% (n = 53)
Gamma Knife Radiosurgery (%)	6.6% (n = 9)	7.7% (n = 4)	6.1% (n = 5)	10.0% (n = 2)	6.7% (n = 4)	3.6% (n = 2)

## Data Availability

The data presented in this study are available on request from the corresponding author.

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
