# Peer review of "Clinical Presentation, Treatment Outcomes, and Demographic Trends in Vestibular Schwannomas: A 135-Case Retrospective Study"

_jcm, 2025, doi:10.3390/jcm14020482_

Round 1
Reviewer 1 Report
Comments and Suggestions for Authors
Dear authors, dear Editor;
The present study study aimed to explore the clinical presentations, treatment outcomes, and potential risk factors of vestibular schwannomas (VS) in a cohort of 135 patients treated between 2006 and 2022. The study provided a comprehensive analysis of patient demographics, symptoms, treatment modalities, and outcomes, with a primary focus on the prevalence of symptoms such as hearing impairment, facial palsy, and balance disorders. The main contributions include a detailed description of the treatment modalities and their outcomes, including a significant focus on the gender disparity in VS cases.
Major comments:
- Data availability statement is missing.
- Figure 1 is redundant and not necessary as it provides no additional information.
- You mentionend hearing impairment as clinical symptom, however, there are no data on the degree of hearing impairment measured by pure-tone audiometry. Moreover, how was the hearing function after the surgery?
- Did you measure vestibular function before and after surgery?
- Data regarding intra- and post-operative complications are missing, Please provide.
Reviewer 2 Report
Comments and Suggestions for Authors
The problem of diagnosis and treatment of vestibular schwannomas has remained relevant for many years, which explains the potential interest of readers in the results of the authors' study. However, the manuscript needs a serious revision.
Major comments
The scientific novelty of the present study is low, since the clinical symptoms of this disease are well known and demonstrated in many previous studies. In addition, the surgical treatment methods analyzed by the authors, including the Gamma Knife, are also well known and have been sufficiently studied. In the Introduction section, the authors need to present the existing gaps in knowledge about vestibular schwannomas in the world in general and in Romania in particular, which prompted them to conduct this study. I recommend that the authors provide in this section a brief summary of the existing problems of surgical treatment, including long-term outcomes. Have similar studies been conducted in Romania?
Neurofibromatosis type 2 is the leading monogenic hereditary disease in which tumors of the Schwann sheath of the vestibular nerve develop and recur in humans. What is the incidence of neurofibromatosis type 2 in Romania and is it different from that in other European countries?
There are no comparison groups (for example, male and female, AND/OR young age, middle age, old age).
Minor comments
Materials and Methods:
Add criteria for the inclusion and exclusion of participants in this study, a flowchart of the study design.
Use the MPI template to design the names of sections and subsections.
Results:
Figure 1 and Figure 3 duplicate the content in the text, its informativeness is low. I recommend removing these figures.
Modify the title of Figure 2 to match the title of your article.
The quality of Figures 4, 5, 6, 7 and 8 is very low. I recommend modifying its into a tables where you can provide information about the presence and severity of symptoms in the general sample, as well as in female and male patients separately.
Obesity, arterial hypertension and diabetes mellitus are comorbid diseases, but not risk factors for vestibular schwannoma. If the authors want to present these results, it is not important to explain the purpose of their study. In addition, the subsections 3.3.1, 3.3.2 and 3.3.3. are very small and uninformative, so it is better to combine them into one section.
Double-check the use of abbreviations. If the abbreviation has already been used and explained before, then use this abbreviation, not the full name.
Do not use abbreviations in the names of tables and figures.
The references need serious revision (please use the MDPI template).
Comments on the Quality of English LanguageI recommend that the authors improve the style of specialized medical English.
Round 2
Reviewer 1 Report
Comments and Suggestions for Authors
My Comments has been adressed accordingly.
Author Response
Dear Reviewer,
Thank you for your thoughtful and detailed feedback on our manuscript. We sincerely appreciate the time and effort you have taken to review our work and provide valuable insights.
Kind regards,
The authors
Reviewer 2 Report
Comments and Suggestions for Authors
In general, the novelty and originality of this article are low.
The authors have modified and improved their manuscript. However, it still needs technical improvement.
The quality of the figures is low, I recommend increasing the font size on the diagrams to improve their readability.
When designing tables, it is necessary to use the MDPI template.
The authors used only basic statistics. The manuscript looks more like a case report rather than an article.
When using the median, add an interquartile interval.
When using the mean, add the standard deviation.
Comments on the Quality of English LanguageThe style of the English language needs correction.
